# Identification of Game Periods and Playing Position Activity Profiles in Elite-Level Beach Soccer Players Through Principal Component Analysis

**DOI:** 10.3390/s24237708

**Published:** 2024-12-02

**Authors:** Pau Vaccaro Benet, Alexis Ugalde-Ramírez, Carlos D. Gómez-Carmona, José Pino-Ortega, Boryi A. Becerra-Patiño

**Affiliations:** 1Department of Physical Activity and Sport, University of Murcia, 30720 Murcia, Spain; josepinoortega@um.es; 2School of Human Movement Science and Quality of Life, Universidad Nacional, Heredia 86-3000, Costa Rica; jose.ugalde.ramirez@una.cr; 3BioVetMed & SportSci Research Group, University of Murcia, 30100 Murcia, Spain; carlosdavid.gomez@um.es; 4Research Group in Optimization of Training and Sports Performance (GOERD), University of Extremadura, 10003 Caceres, Spain; 5Research Group in Training, Physical Activity and Sports Performance (ENFYRED), Department of Music, Plastic and Body Expression, Faculty of Human and Social Sciences, University of Zaragoza, 44003 Teruel, Spain; 6Management and Pedagogy of Physical Activity and Sport (GPAFD), Universidad Pedagógica Nacional, Bogotá 110221, Colombia; babecerrap@pedagogica.edu.co

**Keywords:** performance analysis, load quantification, inertial movement units, playing position, game periods, beach soccer

## Abstract

Beach soccer has gained increasing interest for study in the sports sciences. In this sense, the analysis of activity profiles is important for training design and load individualization. Therefore, the aims of this study were to identify the most important variables to assess the activity profile and to compare them according to the playing position and game periods in international beach soccer matches. A total of 19 matches of the Spanish national beach soccer team were analyzed during their participation in different international competitions during the 2021–2022 season. A Principal Component Analysis (PCA) was applied to objectively select the external load variables that best explain the data. Kaiser–Meyer–Olkin values of 0.705 and Bartlett’s Sphericity (*p* < 0.01) were obtained. Kruskal–Wallis and Friedman tests was performed for playing positions and game period comparisons, respectively. The PCA showed seven components that grouped a total of 20 variables, explaining 66% of the total variance. Only PC1 and PC2 explained more than 15% (23% and 17%, respectively). Differences were found between playing positions (H > 22.73; *p* < 0.01) and between game periods (X2 > 16.46; *p* < 0.01). A significant decrease was found throughout the game, with the highest demands in period 1 and the lowest in period 3. The highest workloads were performed by wingers and the lowest by goalkeepers. The differences between positions and game periods were found in the following: Total Distance (m/min), HIBD (m/min), High Acc (m/s), High Dec (m/s), Acc 1–2 m/s^2^ (m), Acc 2–3 m/s^2^ (m), Imp 4–5G (n), Imp 5–6G (n), Sprint (n), and Dec 10–6 m/s^2^ (m) (*p* < 0.01). In conclusion, physical and tactical demands faced by elite-level beach soccer players will be influenced by playing positions and game periods. Coaches needs to develop position-specific training programs and optimize substitution strategies for enhancing overall team performance.

## 1. Introduction

Beach soccer is a variant of traditional soccer played on sand that has experienced remarkable growth in popularity and competitiveness at all levels over the past two decades [1]. Unlike the traditional 11-a-side game, beach soccer is played with fewer players (five per team) on a smaller pitch (35–37 m in length and 26–28 m in width) and in a sand surface, resulting in a more intense and physically demanding game [2,3]. The distinctive playing surface and rules in beach soccer pose unique physiological and tactical challenges for players compared to the conventional game [4,5].

One of the critical factors influencing performance in beach soccer is the activity profile and workload demands imposed on players during matches [6]. Thus, in beach sports, there are micro-technologies that allow for a time–motion analysis with the aim of quantifying workload through sport specificity [6,7]. The US government’s global navigation systems (GPSs) has been the most widely used technology for external workload monitoring [8]. Subsequently, local positioning systems (LPSs) have favored the measurement of different variables through the GNSS signal and antennas around the court, especially in those environments where satellite coverage is poor [9]. Additionally, due to the specificity of beach soccer and the conditions in which it is practiced, including the surface, the number of players and the actions performed, the use of micro-technologies (accelerometers, gyroscopes, and magnetometers) has been incorporated, which have allowed monitoring of the workload in high-intensity actions without locomotion, such as, for example, jumps, turns, and collisions [10].

Several studies have investigated the physical and physiological responses of beach soccer players, reporting a relative distance covered of 100 m/min, high heart rate values (>90% of heart rate max), and blood lactate concentrations, indicative of the intermittent and high-intensity nature of the sport with a work/rest ratio of 4:1 [1,11,12]. These demands are a result of physical actions, such as accelerations, decelerations or jumps, technical actions such as pass or shots, and the instability of the sand surface [13]. However, these studies have primarily focused on overall match demands, without considering the potential variations in activity profiles across different game periods or playing positions.

Understanding the activity profiles of players during specific game periods and across different playing positions is crucial for developing position-specific training programs, optimizing substitution strategies, and enhancing the overall team performance [14,15,16]. In beach soccer, matches are divided into three periods of 12 min each, with unlimited substitutions allowed [17]. This structure, combined with the physically demanding nature of the sport, may lead to fluctuations in the activity profiles. In this sense, two previous studies that evaluated internal load found lower activity at 86–95% of maximum heart rate and higher blood lactate throughout the match periods due to accumulated fatigue [1,18]. No previous study evaluated the effect of periods in the external workload.

Furthermore, the unique tactical roles and responsibilities associated with different playing positions influence the activity profile. Several studies have investigated the activity profiles based on playing positions in various team sports such as soccer [19] and rugby [20]. However, limited research has been conducted to examine these factors in the context of beach soccer, where different playing positions (e.g., goalkeeper, defender, winger, pivot) may result in distinct activity profiles. Only a previous study compared the external workload demands between playing positions, identifying that the goalkeeper covered less distance as speed >7 km/h. In addition, no differences were found between pivots and goalkeepers in distance covered >18 km/h, and the number of sprints due to the pivot showed the most advanced player on the field and performed numerous duels in a limited space [21].

Accurately quantifying both internal (e.g., heart rate, blood lactate) and external (e.g., distance covered, accelerations) workload demands is crucial for understanding the activity profiles of players throughout the match. Modern technologies, such as global navigation satellite systems (GNSSs) and local positioning systems (LPSs) have enabled the precise tracking of players’ movements and physical demands in team sports in indoor and outdoor conditions [22]. The GNSS and LPS provide valuable data on variables like total distance covered, high-speed running, accelerations, and decelerations, which can be used to characterize the external workload demands of players during matches [23]. Complementing these external load measures with internal load indicators like heart rate and blood lactate concentrations can provide a comprehensive understanding of the physiological strain experienced by players [24].

To reduce the high number of variables registered by technological tools, Principal Component Analysis (PCA) is a multivariate statistical technique widely used in sports science research to identify and interpret patterns in complex datasets [25]. By reducing the dimensionality of the data and extracting principal components (PCs) that capture the majority of the variance, a PCA can provide valuable insights into the underlying structure and relationships among multiple variables [26]. In the context of team sports, the PCA has been successfully applied to analyze activity profiles, differentiate between playing positions, and identify key performance indicators [27,28,29,30].

Therefore, the aims of the present study were as follows: (i) identify the key performance indicators of activity profiles, (ii) determine the existence of distinct activity profiles across different game periods, and (iii) investigate the differences in activity profiles among playing positions in elite-level beach soccer. By addressing these objectives, this study will provide valuable insights into the physical and tactical demands faced by elite-level beach soccer players, enabling coaches and practitioners to develop more targeted and position-specific training programs, optimize substitution strategies, and enhance overall team performance. We hypothesized that a PCA could reduce the big data of external load to 20–30 most important metrics, and we hypothesized that playing positions and game periods could influence the external workload demands during competition in elite-level beach soccer players.

## 2. Materials and Methods

### 2.1. Design

An observational longitudinal study was conducted over the time period of 2021–2022, where a total of 19 official matches played by the Spanish national beach soccer team were registered [31]. All players on the team roster wore portable inertial measurement units (IMUs) positioned at scapulae during matches to measure external workload variables. Following data collection, a PCA was performed on the dataset to identify the key external load variables accounting for the majority of variance. The effects of the playing position (goalkeeper, defender, winger, and pivot) and game period (first 12 min, second 12 min, and third 12 min) on these key variables were evaluated.

### 2.2. Participants

Twenty-two elite-level beach soccer players voluntarily participated in this study (age: 27.5 ± 3.2 years, height: 180 ± 6 cm, body mass: 75 ± 8 kg, and Spanish national team appearances: 25 ± 10 times). All participants possessed significant experience in elite-level beach soccer, averaging 7.34 ± 2.75 years, and, with the Spanish national team, averaging 3.29 ± 1.89 years. The inclusion criteria were as follows: (a) participation in a European or international championship within the past two years, (b) a minimum of four years of competitive beach soccer experience, and (c) the absence of injuries or medical conditions that could impede regular practice of beach soccer. Informed consent was obtained from all participants prior to their inclusion, ensuring a comprehensive understanding of the study’s nature, procedures, and potential risks. This study was conducted in accordance with the ethical principles outlined in the Declaration of Helsinki [32]. This study received approval from the institute’s research ethics committee (ID: 3495/2021) and adhered to the ethical recommendations for research involving human participants, as outlined in the Declaration of Helsinki (2013).

### 2.3. Instruments and Variables

In the present study, the WIMU PRO devices (RealTrack Systems, Almeria, Spain) were used for monitoring and analyzing the physical performance of athletes. It is equipped with tri-axial accelerometers, gyroscopes, and magnetometers, all of which record data at a frequency of 100 Hz, providing high-resolution insights into the athletes’ movements. Additionally, the device features a global positioning system (GPS) that operates at a frequency of 10 Hz, ensuring precise tracking of the athletes’ spatial dynamics during training and competition. The reliability and validity of the WIMU PRO’s accelerometers and GPS sensors were proved previously with excellent results [33,34].

From the sensors that compose the WIMU PRO device, more than 200 metrics can be obtained regarding distance covered at different speeds, accelerations, and decelerations at different intensities, changes in directions at different turning grades, impacts at different intensities, jumps, and step or player loads, among others, for a comprehensive assessment of athletic performance.

### 2.4. Procedures

The data collection for this study was conducted using WIMU PRO devices (RealTrack Systems, Almeria, Spain) during various European and international championships in the 2021–2022 season (e.g., Beach Soccer World Cup, Europe Qualifiers, Euro Beach Soccer League, Intercontinental Beach Soccer Cup). The monitoring of external loads during matches is a routine activity for the team, and no phase of the established process was altered or modified for this study. The inertial devices were affixed to the players 30 min before the start of the game, using a neoprene vest that fits comfortably to the player’s body. The vest has a compartment positioned at the T2-T4 vertebrae level to place the IMU. These vests and devices were worn underneath the team’s uniform. Each athlete tested had their own device.

Each match in the tournaments consisted of three periods of 12 min each. Any overtime periods were not considered for the analysis. Players’ data were included if they played a minimum of 8 min per period (66%), ensuring sufficient data for reliable analysis. After each game, all data were downloaded and processed using the SPRO software 2.2.0 (RealTrack Systems, Almería, Spain). The data were normalized relative to each player’s actual playing time. In total, 474 observations of each variable were considered for analysis.

### 2.5. Statistical Analysis

A Principal Component Analysis (PCA) was conducted to objectively select the external load variables that best explain the data. Initially, Pearson correlations were applied to identify variables with correlation values r < 0.7. After filtering and selecting the variables, they were converted into Z-scores. The Kaiser–Meyer–Olkin (KMO) value obtained was 0.705, and Bartlett’s Test of Sphericity was significant (*p* < 0.01). Components were selected based on eigenvalues ≥1. An orthogonal rotation (Varimax method) was performed to ensure that each principal component provided distinct information. A loading value of ≥0.6 was used to select the variables that constituted each component. Subsequent to the identification of variables, Kolmogorov–Smirnov tests were applied to evaluate normality, which showed values <0.01, indicating that the data did not follow a normal distribution. Therefore, non-parametric tests were employed for analysis. The Kruskal–Wallis test was used for comparisons between playing positions and match outcomes, while the Friedman test was applied for comparisons between game periods. Specific differences were identified using Bonferroni-adjusted post hoc tests. Statistical analyses were performed using the Statistical Package for Social Science (version 24, IBM Corp., Armonk, NY, USA). The significance level was established as *p* < 0.05.

## 3. Results

### 3.1. Principal Component Analysis

After the discrimination of the variables, the PCA revealed seven components, grouping a total of 20 variables and explaining 66% of the total variance (Table 1). To facilitate the exercise, we worked with the first two dimensions (eigenvalues greater than 1) with a cumulative inertia of 40%. This allows us to understand the nature of the variables by the amount of information they provide. Likewise, the other dimensions were not considered because the % variance was not greater than 10.

The variables constituting the first component (1) were as follows: total distance, high-intensity burst distance (HIBD), the time in milliseconds of high accelerations (>3 m/s^2^) and high decelerations (>−3 m/s^2^), and the distance in accelerations of 1–2 m/s^2^ and 2–3 m/s^2^. The second component (2) included the number of accelerations, the number of high accelerations (>3 m/s^2^) and decelerations (>−3 m/s^2^), the distance in decelerations (>−3 m/s^2^), and the medio-lateral Player Load.

### 3.2. Playing Positions

Comparative analyses among playing positions revealed differences in 17 out of the 20 variables analyzed (Table 2). Goalkeepers recorded lower values in their variables compared to other positions, except for the number of accelerations, distance at 0–3 km/h, acceleration density, and distance in accelerations (0–1 m/s^2^), where their values were higher.

Wingers registered higher values compared to other positions in the following variables: total distance, high-intensity burst distance (HIBD), milliseconds of high-intensity accelerations and decelerations, distance in accelerations (2–3 m/s^2^), number of accelerations, the number of high-intensity accelerations and decelerations, and the number of sprints (Table 2).

### 3.3. Game Periods

Differences were found among game periods in 14 out of the 20 variables analyzed. Specifically, variables such as total distance, HIBD, milliseconds in high-intensity accelerations and decelerations, the number of high-intensity decelerations, and acceleration density showed a decrease as the game progressed (1st half > 2nd half > 3rd half) (Table 3). The quantity of high-intensity accelerations was greater in the first half compared to the third.

For variables like distance in accelerations (1–2 m/s^2^, 2–3 m/s^2^), distance in high-intensity decelerations, the average lateral Player Load, and the number of impacts at 4–5 g, the first period was higher than the second and third periods. No differences were found between the second and third periods. The number of accelerations increased in the second period compared to the first, and in the third period compared to the second. The distance covered at 0–3 km/h increased in the third period compared to the first.

## 4. Discussion

Beach soccer has gained increasing popularity in recent years, leading to a growing interest in understanding the physical and tactical demands of the sport. The present study aimed to identify key performance indicators of activity profiles in elite beach soccer players and investigate the influence of playing positions and game periods on these profiles. Our findings revealed that a Principal Component Analysis (PCA) could effectively reduce the dimensionality of external load variables, identifying seven components that explained 66% of the total variance. Moreover, significant differences were observed in activity profiles across playing positions and game periods, with wingers exhibiting the highest workloads and a general decline in physical demands as matches progressed.

### 4.1. Principal Component Analysis

The application of a PCA in our study resulted in the identification of seven principal components, comprising 20 variables that collectively explained 66% of the total variance in external load data. Notably, the first two components accounted for 40% of the variance, emphasizing their importance in characterizing the activity profiles of beach soccer players. The variables loading heavily on these components included total distance, high-intensity breaking distance (HIBD), high-intensity accelerations and decelerations, and distances covered at various acceleration thresholds.

These findings are consistent with previous research in other team sports that have employed PCA to identify key performance indicators. For instance, Oliva-Lozano et al. [28] applied a PCA to analyze activity profiles in professional soccer, identifying total distance, high-speed running, and accelerations/decelerations as crucial variables. Similarly, Gløersen et al. [27] used a PCA in cross-country skiing and found that a small number of components could explain a large portion of the variance in performance data. In futsal, a sport with similar characteristics to beach soccer, Ribeiro et al. [35] utilized statistical procedures to determine that high-intensity actions and accelerometer-derived metrics were the most discriminant variables. Our results extend these findings to beach soccer, demonstrating the utility of the PCA in reducing the complexity of external load data while retaining the most informative variables.

The prominence of variables related to high-intensity actions (e.g., HIBD, high-intensity accelerations/decelerations) in the PCA results indicated the intermittent and explosive nature of beach soccer. This aligns with previous research that reported high-intensity efforts and frequent changes in velocity as characteristic features of beach soccer matches [11]. Furthermore, Leite [2] highlighted the importance of explosive actions in beach soccer due to the sand surface, which increases energy expenditure and imposes greater neuromuscular demands compared to firm surfaces.

The identification of these variables through the PCA provides coaches and practitioners with objective criteria for monitoring and evaluating player performance. However, it is crucial to interpret these metrics in the context of the sport’s specific demands. For example, while total distance is a common metric in many team sports, its relevance in beach soccer might be different due to the smaller pitch size and the unlimited substitution rule [1]. Additionally, the inclusion of accelerometer-derived variables (e.g., Player Load) in PCA components reflects the multidimensional nature of external load in beach soccer. These variables capture the mechanical stress experienced by players, which is particularly relevant given the instability of the sand surface [13]. The integration of such variables alongside traditional GPS-derived metrics provides a more comprehensive assessment of the physical demands in beach soccer.

### 4.2. Influence of Playing Positions

Our analysis revealed significant differences in activity profiles across playing positions, with wingers demonstrating the highest workloads in variables such as total distance, HIBD, and high-intensity accelerations/decelerations. In contrast, goalkeepers exhibited the lowest values in most variables, except for the number of accelerations and distance covered at low speeds (0–3 km/h).

These position-specific differences are consistent with the tactical roles and responsibilities of players in beach soccer. Wingers, often required to cover large areas of the pitch and engage in frequent attacking and defensive transitions, naturally accumulate higher physical demands. This finding is supported by research in other team sports, such as soccer [19] and rugby [20], where wide players have been shown to cover greater distances and perform more high-intensity actions compared to central players. In futsal, a small-sided team sport, Spyrou et al. [36] also found that wingers covered more distance at high speeds than other outfield positions. The demanding role of wingers in beach soccer can be attributed to the sport’s tactical characteristics. Leite and Barreira [5] analyzed the offensive play patterns in beach soccer and highlighted the importance of width in creating scoring opportunities. Consequently, wingers are crucial in stretching the opponents’ defense and are involved in numerous high-intensity lateral movements, which is reflected in their elevated physical output.

The unique demands placed on goalkeepers in beach soccer are reflected in their distinct activity profile. Despite lower overall distances and high-intensity actions, goalkeepers exhibited higher acceleration counts and low-speed distances. This may be attributed to the frequent involvement of goalkeepers in build-up play and the unstable sand surface, requiring numerous small adjustments in positioning. These results partially align with a previous study conducted by Borges et al. [21] who found that goalkeepers covered less high-speed distance but did not differ from outfield players in sprint-related metrics. The high number of accelerations performed by goalkeepers, albeit at lower intensities, underscores their active participation in the game beyond traditional shot-stopping duties. In beach soccer, goalkeepers often act as an additional outfield player, contributing to the team’s offensive plays [2]. This tactical aspect, combined with the challenges posed by the sand surface, leads to a unique activity profile characterized by frequent but less intense movements.

Interestingly, our findings revealed that pivots, despite being the most advanced players, did not always exhibit the highest intensity profile. This contrasts with some observations in soccer, where central forwards often cover large distances at high speeds [19]. However, it aligns with the tactical nuances of beach soccer, where pivots often engage in short, explosive actions within confined spaces [21]. Finally, the need for position-specific training programs in beach soccer was emphasized. Coaches should design drills and conditioning exercises that replicate the physical demands of each playing position, ensuring that players are adequately prepared for their roles.

### 4.3. Influence of Match Periods

There are studies that analyze the activity profiles and external load patterns in soccer, showing that, during the competition, the playing positions express a significant drop in the HIR distance, the distance covered in sprint actions and the number of accelerations, becoming pronounced in the 5 min after the 5 min peak period and in the last 5 min of the last period [37]. Other studies have reported that there is a reduction for the variable HIR, sprint and acceleration after the 5 min peak; however, only the reduction in accelerations in the last 5 min of the match turned out to be significant [38]. These findings are related to those of the present study, where it was determined that there is a significant decrease (*p* < 0.01) in variables such as total distance (m/min), HIBD (m/min), high Acc (m/s), high Dec (m/s), acc 1–2 m/s^2^ (m), and acc 2–3 m/s^2^ (m).

The analysis of activity profiles across the three game periods revealed a significant decline in physical output as matches progressed. Variables such as total distance, HIBD, and high-intensity accelerations/decelerations showed a consistent decrease from the first to the third period. This trend suggests the development of fatigue among players, which is a common phenomenon in intermittent team sports, particularly those played on sand and in challenging environmental conditions [1,11]. The observed decline in physical performance aligns with previous research on the physiological responses of beach soccer players. Scarfone et al. [1] lowered activity at high heart rate zones and increased blood lactate levels in later stages of matches, indicative of accumulated fatigue. Our study extends these findings by providing detailed insights into the changes in external load metrics across game periods.

The fatigue experienced by beach soccer players can be attributed to various factors, including the unstable sand surface and environmental conditions. Playing on sand significantly increases energy expenditure compared to firm surfaces, increasing a 1.6-fold energy cost when walking on sand versus a hard surface [39]. Moreover, Gaudino et al. [40] found that sand surfaces induced greater neuromuscular and metabolic loads during football-specific drills compared to grass. These increased physiological demands, combined with high ambient temperatures and solar radiation often present during beach soccer matches, likely contribute to the rapid onset of fatigue [41].

Interestingly, while most high-intensity variables decreased, the number of accelerations and distance covered at low speeds (0–3 km/h) increased in later periods. This shift in activity profile may reflect tactical adaptations made by players and teams as fatigue sets in, with a greater emphasis on maintaining possession and conserving energy through shorter, less intense actions [42]. Similar tactical adjustments have been observed in other team sports such as soccer where players reduce high-intensity efforts and increase low-intensity activities as matches progress [43].

The temporal changes in activity profiles have important implications for player preparation and in-game management. Coaches should implement strategies to mitigate the effects of fatigue, such as high-intensity interval training and repeated sprint ability drills, specifically designed for sand-based activities [41]. Additionally, heat acclimatization protocols and appropriate hydration strategies are crucial for optimizing performance in hot environments [44]. The timing and frequency of substitutions should be carefully considered to maintain high physical outputs throughout the match, as unlimited substitutions in beach soccer provide an opportunity for tactical periodization of players’ efforts [45].

### 4.4. Limitations and Future Research Directions

Despite the novel insights provided by this study into the activity profiles of elite beach soccer players, some limitations should be acknowledged. The analysis was conducted on a single national team, which may limit the generalizability of findings across different playing styles and competitive levels. Additionally, while external load was comprehensively assessed, internal load measures (e.g., heart rate, subjective ratings) were not included, which could provide a more holistic understanding of the physiological demands. Future research should aim to replicate this study with a larger and more diverse sample of teams and competitions. Integrating both external and internal load metrics would offer valuable insights into the relationship between physical outputs and physiological responses in beach soccer. Longitudinal studies tracking changes in activity profiles across a competitive season could inform periodization strategies. Moreover, investigating the interaction between technical–tactical actions and physical demands would enhance our understanding of the multifaceted nature of performance in beach soccer.

## 5. Conclusions and Practical Applications

The present study provides novel insights into the activity profiles of elite beach soccer players by applying a Principal Component Analysis to objectively identify key performance indicators and investigating the influence of playing positions and game periods on these metrics. Our findings highlight the intermittent and high-intensity nature of beach soccer, characterized by frequent accelerations, decelerations, and explosive actions. The results also demonstrate significant variations in activity profiles across playing positions and game periods, emphasizing the need for individualized and periodized training approaches.

From a practical standpoint, these findings have several important implications for coaches and practitioners. Training programs should be tailored to position-specific demands, with wingers requiring additional high-intensity interval training and goalkeepers focusing on quick, short-distance movements. Effective fatigue management through frequent player rotation is crucial, leveraging the unlimited substitution rule. Conditioning should prioritize improving players’ capacity for high-intensity bursts, rapid accelerations/decelerations, and swift changes in direction, utilizing sand-based plyometrics and small-sided games. Regular monitoring of key performance indicators (e.g., total distance, HIBD, high-intensity accelerations/decelerations) during training and matches ensures appropriate loading and early fatigue detection. Lastly, implementing heat acclimatization protocols and hydration strategies is essential for sustaining performance in challenging environmental conditions.

## Figures and Tables

**Table 1 sensors-24-07708-t001:** PCA results of the external workload variables obtained by WIMU PRO.

Eigenvalues	PC1	PC2	PC3	PC4	PC5	PC6	PC7
**% of variance**	23.464	17.221	7.054	5.994	4.923	4.254	3.345
**% accumulated**	23.464	40.685	47.739	53.732	58.656	62.909	66.254
**Total distance (m/min)**	0.751						
**HIBD (m/min)**	0.819						
**High Acc (m/s)**	0.680						
**High Dec (m/s)**	0.821						
**Acc 1–2 m/s^2^ (m)**	0.741						
**Acc 2–3 m/s^2^ (m)**	0.737						
**Accelerations (n/min)**		−0.920					
**High Acc (n)**		0.904					
**High Dec (n)**		0.789					
**High Dec (m)**		0.709					
**Player Load ML (a.u.)**		0.780					
**Imp 4–5G (n)**			0.756				
**Imp 8–9G (n)**			0.768				
**Imp 5–6G (n)**			0.726				
**Sprint (n)**				0.740			
**Sprint Rel (n)**				0.667			
**Distance 0–3 km/h (m/min)**					−0.691		
**Acc Density**					0.702		
**Acc 0–1 m/s^2^ (m)**						0.685	
**Dec 10–6 m/s^2^ (m)**							0.812

**Note**: m: meters; min: minutes; HIBD: high-intensity break distance; Acc: accelerations; Dec: decelerations; m: meters; s: seconds; Imp 4–5G (n): absolute decelerations from 4 to 5 G; Imp 8–9G (n): Absolute decelerations from 8 to 9 G; and Imp 5–6G (n): absolute decelerations from 5 to 6 G.

**Table 2 sensors-24-07708-t002:** Descriptive and inferential analysis of selected external workload variables based on playing positions.

	W (*n* = 111)	P (*n* = 78)	B (*n* = 120)	W-P (*n* = 102)	W-B (*n* = 24)	GK (*n* = 39)	H	*p*	Post Hoc
**Total** **distance (m/min)**	77.7 + 13.3	73.4 ± 14.8	69.7 ± 12.1	72.01 ± 12.06	68.9 ± 13.6	43.4 ± 18.7	91.668	<0.001	Gk < W, B, P, W-B, W-P; W > B, W-P, W-B
**HIBD (m/min)**	4.09 ± 1.23	3.52 ± 1.14	3.34 ± 1.28	3.44 ± 0.92	3.13 ± 1.02	1.22 ± 1.07	111.450	<0.001	Gk < W, P, B, W-P, W-B; W > B, P, W-B, W-P
**High Acc (m/s)**	200.6 ± 848.6	1595.3 ± 921.3	1518.5 ± 1197.2	1692.2 ± 764.6	1790.4 ± 779.1	569.9 ± 424.7	100.765	<0.001	Gk < W, B, P, W-B, W-P; W > B, P
**High Dec (m/s)**	2375.0 ± 1033.0	1909.6 ± 797.5	1762.0 ± 735.4	2019.3 ± 683.5	1792.9 ± 760.8	812.8 ± 1339.2	96.097	<0.001	Gk < W, B, P, W-B, W-P; W > B, P
**Acc 1–2 m/s^2^ (m)**	12.16 ± 3.60	12.53 ± 4.13	11.58 ± 3.43	11.37 ± 2.89	10.02 ± 2.30	5.90 ± 4.14	76.113	<0.001	Gk < W, B, P, W-B, W-P
**Acc 2–3 m/s^2^ (m)**	10.74 ± 3.61	10.33 ± 3.68	8.93 ± 3.24	9.42 ± 3.03	9.15 ± 3.49	3.23 ± 3.92	94.225	<0.001	Gk < W, B, P, W-B, W-P; W > C
**Acc (n/min)**	30.56 ± 2.78	31.22 ± 62.86	32.21 ± 2.49	31.44 ± 2.71	32.06 ± 1.94	36.15 ± 3.43	88.275	<0.001	Gk > W, B, P, W-B, W-P; C > W, P
**High Acc (n)**	1.34 ± 1.07	1.08 ± 1.21	1.19 ± 1.39	1.07 ± 0.88	0.96 ± 0.41	0.43 ± 0.38	75.494	<0.001	Gk < W, B, P, W-B, W-P; A > B, P, W-P, W-B
**High Dec (n)**	1.56 ± 0.60	1.32 ± 0.54	1.15 ± 0.44	1.27 ± 0.43	1.10 ± 0.45	0.58 ± 0.88	104.572	<0.001	Gk < W, B, P, W-B, W-P; A > B, W-P, W-B
**High Dec (m)**	6.30 ± 2.61	4.77 ± 1.90	4.44 ± 1.76	5.06 ± 1.77	4.24 ± 1.89	1.71 ± 2.71	116.084	<0.001	Gk < W, B, P, W-B, W-P; A > B, P, W-P, W-B
**Player Load ML (a.u.)**	0.57 ± 0.41	0.53 ± 0.49	0.49 ± 0.24	0.48 ± 0.16	0.47 ± 0.10	0.29 ± 0.10	86.159	<0.001	Gk < W, B, P, W-B, W-P; A > B, W-P
**Imp 4–5G (n)**	3.57 ± 2.09	2.92 ± 1.59	3.66 ± 2.00	3.83 ± 2.50	4.59 ± 2.06	1.13 ± 0.76	75.955	<0.001	Gk < W, B, P, W-B, W-P; C < W-P
**Imp 8–9G (n)**	0.15 ± 0.20	0.15 ± 0.19	0.16 ± 0.22	0.16 ± 0.17	0.17 ± 0.19	0.08 ± 0.15	10.481	0.063	N/A
**Imp 5–6G (n)**	0.32 ± 0.26	0.32 ± 0.36	0.36 ± 0.33	0.49 ± 0.42	0.42 ± 0.27	0.16 ± 0.15	31.427	<0.001	Gk < W, B, W-B, W-P; P < W-P
**Sprint (n)**	0.07 ± 0.10	0.02 ± 0.06	0.06 ± 0.11	0.04 ± 0.07	0.01 ± 0.03	0.01 ± 0.04	40.318	<0.001	Gk < W, B, W-P; A > P, W-B
**Sprint Rel (n)**	0.01 ± 0.06	0.02 ± 0.16	0.05 ± 0.23	0.02 ± 0.16	0.00 ± 0.00	0.02 ± 0.09	8.333	0.139	N/A
**Distance 0–3 km/h (m/min)**	9.19 ± 2.26	9.43 ± 1.82	10.72 ± 2.03	9.48 ± 1.94	11.02 ± 2.04	12.18 ± 4.32	83.944	<0.001	Gk > W, P, W-P; W-B > W, B, W-P; C > W-P
**Acc Density**	0.11 ± 0.13	0.11 ± 0.15	0.15 ± 0.29	0.11 ± 0.19	0.07 ± 0.03	0.18 ± 0.42	22.732	<0.001	Gk > W, P
**Acc 0–1 m/s^2^ (m)**	12.63 ± 1.93	12.53 ± 1.82	12.85 ± 2.23	13.03 ± 2.06	12.65 ± 1.51	13.56 ± 8.11	12.088	0.034	Gk > W-P
**Dec 10–6 m/s^2^ (m)**	0.20 ± 0.46	0.10 ± 0.29	0.11 ± 0.32	0.14 ± 0.29	0.06 ± 0.16	0.02 ± 0.07	9.540	0.089	N/A

**Note.** GK: goalkeeper, W: wing, B: back, P: pivot, W-P: wing-pivot; W-B: wing-back; m: meters; min: minutes; HIBD: High-intensity break distance; Acc: Accelerations; Dec: Decelerations; m: meters; s: seconds; Imp 4-5G (n): Absolute decelerations from 4 to 5 G; Imp 8-9G (n): Absolute decelerations from 8 to 9 G; Imp 5-6G (n): Absolute decelerations from 5 to 6 G; N/A: Not applicable.

**Table 3 sensors-24-07708-t003:** Descriptive and inferential analysis of selected external workload variables based on match periods.

	P1 (*n* = 158)	P2 (*n* = 158)	P3 (*n* = 158)	Total (*n* = 474)	X2	*p*	Post Hoc
**Total distance (m/min)**	77.25 ± 15.35	69.00 ± 14.91	65.28 ± 15.56	70.51 ± 16.05	61.532	<0.001	1T > 2T > 3T
**HIBD (m/min)**	3.74 ± 1.33	3.31 ± 1.20	3.09 ± 1.42	3.38 ± 1.34	48.973	<0.001	1T > 2T > 3T
**High Acc (m/s)**	1847.4 ± 1045.0	1598.6 ± 837.7	1411.0 ± 1011.2	1619.0 ± 983.2	21.657	<0.001	1T > 2T > 3T
**High Dec (m/s)**	2179.6 ± 1068.0	1893.1 ± 896.1	1653.3 ± 827.2	1908.7 ± 958.4	25.587	<0.001	1T > 2T > 3T
**Acc 1–2 m/s^2^ (m)**	12.38 ± 3.86	10.88 ± 3.63	10.58 ± 3.93	11.28 ± 3.88	17.633	<0.001	1T > 2T; 1T > 3T
**Acc 2–3 m/s^2^ (m)**	10.29 ± 4.27	9.00 ± 3.71	8.41 ± 3.54	9.23 ± 3.92	36.000	<0.001	1T > 2T; 1T > 3T
**Accelerations (n/min)**	30.70 ± 2.75	32.17 ± 3.04	32.57 ± 3.11	31.81 ± 3.07	43.152	<0.001	1T < 2T; 1T < 3T
**High Acc (n)**	1.25 ± 1.19	1.08 ± 1.01	1.00 ± 1.12	1.11 ± 1.11	17.943	<0.001	1T > 3T
**High Dec (n)**	1.44 ± 0.68	1.23 ± 0.53	1.09 ± 0.52	1.25 ± 0.60	37.181	<0.001	1T > 2T > 3T
**High Dec (m)**	5.47 ± 2.64	4.77 ± 2.31	4.24 ± 2.08	4.83 ± 2.40	21.827	<0.001	1T > 2T; 1T > 3T
**Player Load ML (a.u.)**	0.52 ± 0.21	0.48 ± 0.29	0.49 ± 0.43	0.50 ± 0.32	70.228	<0.001	1T > 2T; 1T > 3T
**Imp 4–5G (n)**	3.73 ± 62.40	3.30 ± 2.03	3.15 ± 1.96	3.39 ± 2.15	16.456	<0.001	1T > 2T; 1T > 3T
**Imp 8–9G (n)**	0.16 ± 0.22	0.16 ± 0.19	0.13 ± 0.17	0.15 ± 0.19	5.004	0.082	N/A
**Imp 5–6G (n)**	0.37 ± 0.35	0.35 ± 0.31	0.35 ± 0.35	0.36 ± 0.34	0.465	0.793	N/A
**Sprint (n)**	0.05 ± 0.10	0.05 ± 0.09	0.03 ± 0.06	0.05 ± 0.09	5.259	0.072	N/A
**Sprint Rel (n)**	0.02 ± 0.16	0.04 ± 0.21	0.01 ± 0.08	0.03 ± 0.16	2.889	0.236	N/A
**Distance 0–3 km/h (m/min)**	9.74 ± 2.24	10.05 ± 2.26	10.27 ± 2.86	10.02 ± 2.47	7.452	0.024	1T < 3T
**Acc Density**	0.17 ± 0.30	0.10 ± 0.15	0.10 ± 0.20	0.12 ± 0.23	56.903	<0.001	1T > 2T > 3T
**Acc 0–1 m/s^2^ (m)**	13.29 ± 4.26	12.77 ± 2.03	12.44 ± 2.14	12.83 ± 3.00	1.861	0.394	N/A
**Dec 10–6 m/s^2^ (m)**	0.13 ± 0.38	0.15 ± 0.35	0.10 ± 0.26	0.13 ± 0.33	0.826	0.662	N/A

**Note.** P1: Period 1, P2: Period 2, P3: Period 3; m: meters; min: minutes; HIBD: High-intensity break distance; Acc: Accelerations; Dec: Decelerations; m: meters; s: seconds; Imp 4–5G (n): Absolute decelerations from 4 to 5 G; Imp 8–9G (n): Absolute decelerations from 8 to 9 G; Imp 5–6G (n): Absolute decelerations from 5 to 6 G; N/A: Not applicable.

## Data Availability

Data are contained within the article.

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
