# Peer review of "Identification of Game Periods and Playing Position Activity Profiles in Elite-Level Beach Soccer Players Through Principal Component Analysis"

_sensors, 2024, doi:10.3390/s24237708_

Round 1
Reviewer 1 Report
Comments and Suggestions for Authors
This manuscript reports on a study that measures the movements of beach soccer players during matches using accelerometers. It has the potential to contribute to the development of beach soccer, and I would like to offer the following comments from an academic perspective, hoping the authors will consider them.
#1
The background provides limited mention of the significance of using accelerometers, which may cause readers to mistakenly believe the study focuses on heart rate or blood lactate measurements. I recommend elaborating more in the background on how the use of accelerometers adds academic value to the study.
#2
There should be more detailed mention of considerations when using accelerometers in the specific context of beach soccer. For example, the characteristics of the surface in beach soccer (in comparison to regular soccer) and the fact that there are fewer player positions than in soccer should be taken into account when assessing whether the sample size is appropriate for position-based comparisons.
#3
While PCA is suitable for dimensionality reduction, it requires more explanation. The PCA identified seven components, explaining 66% of the variance. It would be beneficial to provide a more detailed discussion of why the first two components are considered particularly important and how the other components contribute to the understanding of player workloads (and why the first two components are emphasized while the others are not discussed as much).
#4
The trend of declining physical performance as the match progresses is noted, but it would be beneficial to examine the extent to which these findings align with other research, particularly studies on beach soccer or traditional soccer.
#5
In terms of practical applications, the discussion of player training and substitution strategies is relevant. However, it would be more useful if there were further explanation of how the findings specifically influence tactical decision-making during matches.
Author Response
Dear Reviewer,
Thank you for reviewing our manuscript at a second time. As a research group we value your effort and input.
We have followed your suggestions point by point to improve the manuscript quality, according to our possibilities. The changes have been made in the full text using the red color so that you can see them. Thanks for your time. Once again, we thank you for your valuable contributions, which have helped to strengthen the document.
Comments 1. The background provides limited mention of the significance of using accelerometers, which may cause readers to mistakenly believe the study focuses on heart rate or blood lactate measurements. I recommend elaborating more in the background on how the use of accelerometers adds academic value to the study.
Comments 2. There should be more detailed mention of considerations when using accelerometers in the specific context of beach soccer. For example, the characteristics of the surface in beach soccer (in comparison to regular soccer) and the fact that there are fewer player positions than in soccer should be taken into account when assessing whether the sample size is appropriate for position-based comparisons.
Response 1 and 2. The changes suggested by the reviewer were accepted (Line 49-60).
Thus, in beach sports there are micro technologies that allow time-motion analysis with the aim of quantifying workload through sport specificity [6,7]. The US government's global navigation systems (GPS) has been the most widely used technology for external workload monitoring [8]. Subsequently, local positioning systems (LPS) have favored the measurement of different variables through the GNSS signal and antennas around the court, especially in those environments where satellite coverage is poor [9]. Additionally, due to the specificity of beach soccer and the conditions in which it is practiced, including the surface, the number of players and the actions performed, the use of micro technologies (accelerometers, gyroscopes and magnetometers) has been incorporated, which have allowed monitoring of the workload in high intensity actions without locomotion, such as, for example, jumps, turns and collisions [10].
Comments 3. While PCA is suitable for dimensionality reduction, it requires more explanation. The PCA identified seven components, explaining 66% of the variance. It would be beneficial to provide a more detailed discussion of why the first two components are considered particularly important and how the other components contribute to the understanding of player workloads (and why the first two components are emphasized while the others are not discussed as much).
Response 3. The changes suggested by the reviewer were accepted. (Line 194-199).
After discrimination of the variables, Principal Component Analysis (PCA) revealed seven components grouping a total of 20 variables and explaining 66% of the total variance (Table 1). To facilitate the exercise, we worked with the first two dimensions (eigen-values greater than 1) with a cumulative inertia of 40%. This allows us to understand the nature of the variables by the amount of information they provide. Likewise, the other dimensions were not considered because the % variance was not greater than 10.
Comments 4. The trend of declining physical performance as the match progresses is noted, but it would be beneficial to examine the extent to which these findings align with other research, particularly studies on beach soccer or traditional soccer.
Response 4. The changes suggested by the reviewer were accepted. (Line 317-326).
There are studies that analyze the activity profiles and external load patterns in soccer, showing that during the competition the playing positions express a significant drop in the HIR distance, distance covered in sprint actions and number of accelerations, be-coming pronounced in the 5 minutes after the 5-minute peak period and in the last 5 minutes of the last period [37]. Other studies have reported that there is a reduction for the variable HIR, sprint and acceleration after the 5-minute peak, however, only the reduction of accelerations in the last 5 minutes of the match turned out to be significant [38]. These findings are related to those of the present study, where it was determined that there is a significant decrease (p<0.01) in variables such as total distance (m/min), HIBD (m/min), high Acc (m/s), high Dec (m/s), acc 1-2 m/s² (m) and acc 2-3 m/s² (m).
Comments 5. In terms of practical applications, the discussion of player training and substitution strategies is relevant. However, it would be more useful if there were further explanation of how the findings specifically influence tactical decision-making during matches.
Response 5. We consider that the study is focused on the identification of the activity profile in response to fatigue that can be caused by the demands of the game throughout the match and the needs of each playing position. Therefore, the practical applications are focused on what could be evaluated. It was added the possibility of considering in future studies the integration of the decision-making variable or evaluation of sport tactics. (Line 372-376).
From a practical standpoint, these findings have several important implications for coaches and practitioners. Training programs should be tailored to position-specific demands, with wingers requiring additional high-intensity interval training and goalkeepers focusing on quick, short-distance movements. Effective fatigue management through frequent player rotation is crucial, leveraging the unlimited substitution rule. Conditioning should prioritize improving players' capacity for high-intensity bursts, rapid accelerations/decelerations, and swift changes of direction, utilizing sand-based plyometrics and small-sided games. Regular monitoring of key performance indicators (e.g., total distance, HIBD, high-intensity accelerations/decelerations) during training and matches ensures appropriate loading and early fatigue detection. Lastly, implementing heat acclimatization protocols and hydration strategies is essential for sustaining performance in challenging environmental conditions.
Thank you for your positive feedback on our research. Your valuable suggestions greatly contributed to the improvement of our work.
Best regards
Reviewer 2 Report
Comments and Suggestions for Authors
This paper aims to identify the game periods and playing position activity by PCA. The topic is interesting but there are some questions for the authors:
1. Why the PCA was applied? It should be explained clearly since PCA is a traditional method that has been widely used for decades.
2. The data were collected from portable inertial measurement units (IMUs) positioned at scapulae during matches as mentioned in the manuscript. But what the specific information about IMUs such as its company, how many IMUs were used and how they are attached to the human boy?
3. The specific words of abbreviation “PCA” illustrated at the first time is enough, so delete the corresponding explanation in line 114 and 183.
4. Please check the manuscript carefully and correct some errors such as “statiscal” in the manuscript.
5. The description of “statistical analysis” from line 165 to line 180 should be improved since it is confusing.
6. What are the contributions and limitations of this study? Please add corresponding illustration.
7. If possible, could authors added some figures of the results to improve the readability of this paper?
Author Response
Dear Reviewer,
Thank you for reviewing our manuscript at a second time. As a research group we value your effort and input.
We have followed your suggestions point by point to improve the manuscript quality, according to our possibilities. The changes have been made in the full text using the red color so that you can see them. Thanks for your time. Once again, we thank you for your valuable contributions, which have helped to strengthen the document.
Comments 1. Why the PCA was applied? It should be explained clearly since PCA is a traditional method that has been widely used for decades.
Response 1. The changes suggested by the reviewer were accepted. It explains that principal component analysis is a technique used to describe a data set in terms of new, uncorrelated variables (“components”) that allow the information to be viewed from different perspectives. For this reason, two dimensions were chosen as they were the most representative and provided the greatest amount of information (40% of the total inertia).
After discrimination of the variables, Principal Component Analysis (PCA) revealed seven components grouping a total of 20 variables and explaining 66% of the total variance (Table 1). To facilitate the exercise, we worked with the first two dimensions (eigen-values greater than 1) with a cumulative inertia of 40%. This allows us to understand the nature of the variables by the amount of information they provide. Likewise, the other dimensions were not considered because the % variance was not greater than 10.
The variables constituting the first component (1) were: total distance, high intensity burst distance (HIBD), time in milliseconds of high accelerations (> 3 m/s²) and high decelerations (> -3 m/s²), distance in accelerations of 1-2 m/s² and 2-3 m/s². The second component (2) included the number of accelerations, the number of high accelerations (> 3 m/s²) and decelerations (> -3 m/s²), the distance in decelerations (> -3 m/s²), and the medio-lateral Player Load.
Comments 2. The data were collected from portable inertial measurement units (IMUs) positioned at scapulae during matches as mentioned in the manuscript. But what the specific information about IMUs such as its company, how many IMUs were used and how they are attached to the human boy?
Response 2. The changes suggested by the reviewer were accepted
The vest has a compartment positioned at the T2-T4 vertebrae level to place the IMU. These vests and devices were worn underneath the team's uniform. Each athlete tested had his or her own device.
Comments 3. The specific words of abbreviation “PCA” illustrated at the first time is enough, so delete the corresponding explanation in line 114 and 183.
Response 3. The changes suggested by the reviewer were accepted
Comments 4. Please check the manuscript carefully and correct some errors such as “statiscal” in the manuscript.
Response 4. The changes suggested by the reviewer were accepted. (Line 175).
Comments 5. The description of “statistical analysis” from line 165 to line 180 should be improved since it is confusing.
Response 5. We are grateful for the reviewer's valuable suggestion. In view of this, we consider as a research group that the statistical analysis route as described is correct.
Comments 6. What are the contributions and limitations of this study? Please add corresponding illustration.
Response 6. The changes suggested by the reviewer were accepted. (Line 363-376).
Despite the novel insights provided by this study into the activity profiles of elite beach soccer players, some limitations should be acknowledged. The analysis was con-ducted on a single national team, which may limit the generalizability of findings across different playing styles and competitive levels. Additionally, while external load was comprehensively assessed, internal load measures (e.g., heart rate, subjective ratings) were not included, which could provide a more holistic understanding of the physiological demands. Future research should aim to replicate this study with a larger and more di-verse sample of teams and competitions. Integrating both external and internal load met-rics would offer valuable insights into the relationship between physical outputs and physiological responses in beach soccer. Longitudinal studies tracking changes in activity profiles across a competitive season could inform periodization strategies. Moreover, investigating the interaction between technical-tactical actions and physical demands would enhance our understanding of the multifaceted nature of performance in beach soccer.
Comments 7. If possible, could authors added some figures of the results to improve the readability of this paper?
Response 7. We are grateful for the reviewer's valuable suggestion. In view of this, we consider as a research group that the tables summarize better the information obtained.
Thank you for your positive feedback on our research. Your valuable suggestions greatly contributed to the improvement of our work.
Best regards
Round 2
Reviewer 1 Report
Comments and Suggestions for Authors
Dear Authors,
Thank you for revising the manuscript.
I have reviewed all the revisions made.
The additional descriptions have effectively clarified the points that I felt were insufficiently addressed or unclear during my initial review.
Best regards